# Emergent second law for non-equilibrium steady states

**José Nahuel Freitas** [1,2] ✉ **& Massimiliano Esposito** [1,2] ✉

The Gibbs distribution universally characterizes states of thermal equilibrium. In order to extend the Gibbs distribution to non-equilibrium steady states, one must relate the self-information $\mathcal{I}(x) = -\log(P_{ss}(x))$ of microstate $x$ to measurable physical quantities. This is a central problem in non-equilibrium statistical physics. By considering open systems described by stochastic dynamics which become deterministic in the macroscopic limit, we show that changes $\Delta\mathcal{I} = \mathcal{I}(x_t) - \mathcal{I}(x_0)$ in steady state self-information along deterministic trajectories can be bounded by the macroscopic entropy production $\Sigma$. This bound takes the form of an emergent second law $\Sigma + k_b\Delta\mathcal{I} \geq 0$, which contains the usual second law $\Sigma \geq 0$ as a corollary, and is saturated in the linear regime close to equilibrium. We thus obtain a tighter version of the second law of thermodynamics that provides a link between the deterministic relaxation of a system and the non-equilibrium fluctuations at steady state. In addition to its fundamental value, our result leads to novel methods for computing non-equilibrium distributions, providing a deterministic alternative to Gillespie simulations or spectral methods.

When a system is at equilibrium with its environment (i.e., when no energy currents are exchanged) the probability of a given microstate $\boldsymbol{x}$ is given by the Gibbs distribution[1–3]

$$P_{eq}(\boldsymbol{x}) = e^{-\beta\Phi(\boldsymbol{x})}/Z, \qquad (1)$$

where $\beta = (k_b T)^{-1}$ is the inverse temperature of the environment, $\Phi(\boldsymbol{x})$ is the free energy of microstate $\boldsymbol{x}$ (for states with no internal entropy, $\Phi(\boldsymbol{x})$ is just the energy), and $Z = \sum_{\boldsymbol{x}} \exp(-\beta\Phi(\boldsymbol{x}))$ is the partition function. This central result of equilibrium statistical physics has universal validity and its relevance in most areas of physics cannot be overstated. A natural question is whether or not a similar result also holds for nonequilibrium steady states (NESSs), when the system is maintained out of thermal equilibrium by external drives and subjected to constant flows of energy or matter. In this case, one can always write the steady state distribution over microstates as

$$P_{ss}(\boldsymbol{x}) = e^{-\mathcal{I}(\boldsymbol{x})} \qquad (2)$$

in terms of the self-information $\mathcal{I}(\boldsymbol{x})$, also known as fluctuating entropy[4,5]. In order to provide a useful generalization of the Gibbs distribution to NESSs one must relate the self-information $\mathcal{I}(\boldsymbol{x})$ to measurable physical quantities. This quest has a long history, starting with the seminal contributions of Lebowitz and MacLennan[6–8] and followed by other works[9–15]. However, it still remains an open problem in non-equilibrium statistical physics, since previous formal results are simply not practical in computations, due to the fact that they involve averages over stochastic trajectories.

In this work we prove, for a very general class of open systems displaying a macroscopic limit where a deterministic dynamics

---

[1]Complex Systems and Statistical Mechanics, Department of Physics and Materials Science, University of Luxembourg, 162a, avenue de la Faïencerie, Luxembourg L-1511 Luxembourg, Luxembourg. [2]These authors contributed equally: José Nahuel Freitas, Massimiliano Esposito. ✉e-mail: nahuel.freitas@uni.lu; massimiliano.esposito@uni.lu

emerges, the following fundamental bound on changes of self-information:

$$\Sigma_a \equiv \Sigma + k_b(\mathcal{I}(\boldsymbol{x}_t) - \mathcal{I}(\boldsymbol{x}_0)) \geq 0, \tag{3}$$

where $\Sigma = \int_0^t dt' \dot{\Sigma}(\boldsymbol{x}_{t'})/T$ is the entropy production along a deterministic trajectory from microstate $\boldsymbol{x}_0$ to microstate $\boldsymbol{x}_t$. For example, let us consider the case of chemical reaction networks. The concentrations $\boldsymbol{x} = (x_1, x_2, \cdots)$ of different chemical species reacting in a solution are stochastic quantities, and their evolution is therefore described by a probability distribution $P_t(\boldsymbol{x})$ at time $t$. As the volume $V$ of the solution is increased, the distribution $P_t(\boldsymbol{x})$ becomes strongly localised around the most probable values $\boldsymbol{x}_t$ for the concentrations at time $t$, and these values follow a deterministic dynamics that is in general nonlinear (given in this case by the chemical rate equations). An analogous situation is encountered in electronic circuits, where the state variables $\boldsymbol{x}$ are now the voltages at the nodes of a circuit, and the macroscopic limit corresponds to increasing the typical capacitance $C$ of the nodes (as well as the conductivity of the conduction channels connecting pairs of nodes). The remarkable feature of the result in Eq. (3) is that it provides a link between the deterministic dynamics that emerges in the macroscopic limit and the fluctuations observed at steady state. For example, in an electronic circuit powered by voltage sources and working at temperature $T$, the entropy production rate is $\dot{\Sigma} = -\dot{Q}/T$, where $-\dot{Q}$ is the rate of heat dissipation by the conductive elements of the circuit, that can be easily evaluated at the deterministic level. Then, by Eq. (3), the quantity $-\Sigma/k_b = \int_0^t dt' \dot{Q}(\boldsymbol{x}_{t'})/(k_b T)$ provides a lower bound to the change of steady state self-information $\mathcal{I}(\boldsymbol{x}_t) - \mathcal{I}(\boldsymbol{x}_0)$ along a trajectory. To arrive at our main result in Eq. (3) we consider stochastic systems with a well defined macroscopic limit in which the self-information $\mathcal{I}(x)$ can be shown to be extensive, and Eq. (3) is strictly valid in that limit. However, as we also show, our results can be applied to micro or mesoscopic systems whenever sub-extensive contributions to $\mathcal{I}(x)$ can be neglected. We interpret our result as an emergent second law of thermodynamics, that is stronger than the usual second law $\Sigma \geq 0$. This last inequality is recovered from Eq. (3) by considering the fact that $\Delta \mathcal{I} = \mathcal{I}(\boldsymbol{x}_t) - \mathcal{I}(\boldsymbol{x}_0) \leq 0$ (to dominant order in the macroscopic limit, the steady state self-information is a Lyapunov function of the deterministic dynamics[16]). In addition to its conceptual value, our result offers a practical tool to approximate or bound non-equilibrium distributions, that can typically only be accessed via stochastic numerical methods (for example the Gillespie algorithm). In contrast, Eq. (3) only requires to know the deterministic dynamics of the system, which is directly given by the well known network analysis techniques commonly applied in electronic circuits and chemical reaction networks. Furthermore, the inequality in Eq. (3) is saturated close to equilibrium, leading to a powerful linear response theory[17]. As an example, we apply our results to a realistic model of non-equilibrium electronic memory: the normal implementation of SRAM (static random access memory) cells in CMOS (complementary metal-oxide-semiconductor) technology. These memories have a non-equilibrium phase transition from a monostable phase to a bistable phase that allows the storage of a bit of information. As we will see, the transition is well captured by Eq. (3), which also allows to bound the probability of fluctuations around the deterministic fixed points. Finally, we show that a general coarse-graining procedure generates equivalent models with minimal entropy production, and that in this way the bound in Eq. (3) becomes tighter. When applied to the CMOS memory, this improved bound enables the full reconstruction of the steady state distribution arbitrarily away from equilibrium.

## Results

To obtain Eq. (3) we consider stochastic systems described by autonomous Markov jump processes. Thus, let $\{\boldsymbol{n} \in \mathbb{N}^k\}$ be the set of possible states of the system, and $\lambda_\rho(\boldsymbol{n})$ be the rates at which jumps

$\boldsymbol{n} \rightarrow \boldsymbol{n} + \boldsymbol{\Delta}_\rho$ occur, for $\rho = \pm 1, \pm 2, \cdots$ and $\boldsymbol{\Delta}_{-\rho} = -\boldsymbol{\Delta}_\rho$ ($\rho$ indexes a possible jump and $\boldsymbol{\Delta}_\rho$ is the corresponding change in the state). Each state has energy $E(\boldsymbol{n})$ and internal entropy $S(\boldsymbol{n})$. Thermodynamic consistency is introduced by the local detailed balance (LDB) condition[18,19]. It relates the forward and backward jump rates of a given transition with the associated entropy production:

$$\sigma_\rho = \log \frac{\lambda_\rho(\boldsymbol{n})}{\lambda_{-\rho}(\boldsymbol{n} + \boldsymbol{\Delta}_\rho)} = -\beta\Big[\Phi(\boldsymbol{n} + \boldsymbol{\Delta}_\rho) - \Phi(\boldsymbol{n}) - W_\rho(\boldsymbol{n})\Big]. \tag{4}$$

In the previous equation, $\Phi(\boldsymbol{n}) = E(\boldsymbol{n}) - TS(\boldsymbol{n})$ is the free energy of state $\boldsymbol{n}$, and $W_\rho(\boldsymbol{n})$ is the non-conservative work provided by external sources during the transition. For simplicity, we have considered isothermal conditions at inverse temperature $\beta = (k_b T)^{-1}$, and therefore the system is taken away from equilibrium by the external work sources alone. More general situations in which a system interacts with several reservoirs at different temperatures can be treated in the same way, this time in terms of a Massieu potential taking the place of $\beta\Phi(\boldsymbol{n})$[18]. Important classes of systems accepting the previous description are chemical reaction networks and electronic circuits, which are powered by chemical or electrostatic potential differences, respectively. Note that, by energy conservation, the heat provided by the environment during transition $\boldsymbol{n} \rightarrow \boldsymbol{n} + \boldsymbol{\Delta}_\rho$ is $Q_\rho(\boldsymbol{n}) = E(\boldsymbol{n} + \boldsymbol{\Delta}_\rho) - E(\boldsymbol{n}) - W_\rho(\boldsymbol{n})$, and therefore $k_b\sigma_\rho = -Q_\rho(\boldsymbol{n})/T + S(\boldsymbol{n} + \boldsymbol{\Delta}_\rho) - S(\boldsymbol{n})$.

The probability distribution $P_t(\boldsymbol{n})$ over the states of the system at time $t$ evolves according to the master equation

$$\partial_t P_t(\boldsymbol{n}) = \sum_\rho \Big[\lambda_\rho(\boldsymbol{n} - \boldsymbol{\Delta}_\rho)P_t(\boldsymbol{n} - \boldsymbol{\Delta}_\rho) - \lambda_\rho(\boldsymbol{n})P_t(\boldsymbol{n})\Big]. \tag{5}$$

From the master equation and the LDB conditions one can derive the energy balance

$$d_t \langle E \rangle = \langle \dot{W} \rangle + \langle \dot{Q} \rangle, \tag{6}$$

and the usual version of the second law:

$$\begin{aligned} \dot{\Sigma} &= \dot{\Sigma}_e + d_t \langle S \rangle \\ &= \frac{k_b}{2} \sum_{\rho,\boldsymbol{n}} (j_\rho(\boldsymbol{n}) - j_{-\rho}(\boldsymbol{n} + \boldsymbol{\Delta}_\rho)) \log \frac{j_\rho(\boldsymbol{n})}{j_{-\rho}(\boldsymbol{n} + \boldsymbol{\Delta}_\rho)} \geq 0, \end{aligned} \tag{7}$$

where $j_\rho(\boldsymbol{n}) = \lambda_\rho(\boldsymbol{n})P_t(\boldsymbol{n})$ is the current associated to transition $\rho$. In the previous equations, $\langle S \rangle = \sum_{\boldsymbol{n}} P_t(\boldsymbol{n})(S(\boldsymbol{n}) - k_b \log(P_t(\boldsymbol{n})))$ is the entropy of the system, $\langle E \rangle = \sum_{\boldsymbol{n}} E(\boldsymbol{n})P_t(\boldsymbol{n})$ is the average energy, and $\dot{\Sigma}_e$ is the entropy flow rate, given by

$$T\dot{\Sigma}_e = -\langle \dot{Q} \rangle = -\sum_{\rho,\boldsymbol{n}} Q_\rho(\boldsymbol{n})j_\rho(\boldsymbol{n}) \tag{8}$$

where we also defined the heat rate $\langle \dot{Q} \rangle$ (the work rate $\langle \dot{W} \rangle$ is analogously defined as $\langle \dot{W} \rangle = \sum_{\rho,\boldsymbol{n}} W_\rho(\boldsymbol{n})j_\rho(\boldsymbol{n})$). Finally, Eq. (7) can be also expressed as:

$$T\dot{\Sigma} = -d_t \langle F \rangle + \langle \dot{W} \rangle \geq 0 \tag{9}$$

where $\langle F \rangle = \langle E \rangle - T \langle S \rangle$ is the non-equilibrium free energy.

### Adiabatic/nonadiabtic decomposition

If the support of $P_t(\boldsymbol{n})$ can be restricted to a finite subspace of the state space, the Perron-Frobenius theorem states that the master equation in Eq. (5) has a unique steady state $P_{ss}(\boldsymbol{n})$. Once the steady state is attained, the entropy production rate $\dot{\Sigma}$ matches the entropy flow rate $\dot{\Sigma}_e$. An interesting decomposition of the entropy production rate can be obtained by considering the relative entropy $D = \sum_{\boldsymbol{n}} P_t(\boldsymbol{n}) \log(P_t(\boldsymbol{n})/P_{ss}(\boldsymbol{n}))$ between the instantaneous distribution

$P_t(\boldsymbol{n})$ and the steady state distribution $P_{ss}(\boldsymbol{n})$. Then, it is possible to show that $\dot{\Sigma} = \dot{\Sigma}_a + \dot{\Sigma}_{na}$, where

$$\dot{\Sigma}_a = \frac{k_b}{2} \sum_{\rho,\boldsymbol{n}} (j_\rho(\boldsymbol{n}) - j_{-\rho}(\boldsymbol{n} + \boldsymbol{\Delta}_\rho)) \log \frac{j_\rho^{ss}(\boldsymbol{n})}{j_{-\rho}^{ss}(\boldsymbol{n} + \boldsymbol{\Delta}_\rho)}, \qquad (10)$$

and

$$\begin{aligned} \dot{\Sigma}_{na} &= \frac{k_b}{2} \sum_{\rho,\boldsymbol{n}} (j_\rho(\boldsymbol{n}) - j_{-\rho}(\boldsymbol{n} + \boldsymbol{\Delta}_\rho)) \log \frac{P_t(\boldsymbol{n}) P_{ss}(\boldsymbol{n} + \boldsymbol{\Delta}_\rho)}{P_{ss}(\boldsymbol{n}) P_t(\boldsymbol{n} + \boldsymbol{\Delta}_\rho)} \\ &= -k_b d_t D \end{aligned} \qquad (11)$$

are the adiabatic and non-adiabatic contributions to the entropy production rate $\dot{\Sigma}$, respectively. In Eq. (10) we have introduced the steady state probability currents $j_\rho^{ss}(\boldsymbol{n}) = \lambda_\rho(\boldsymbol{n}) P_{ss}(\boldsymbol{n})$. The non-adiabatic contribution $\dot{\Sigma}_{na}$ is related to the relaxation of the system towards the steady state, since it vanishes when the steady state is reached. This is further evidenced by the identity in the second line of Eq. (11): a reduction in the relative entropy between $P_t(\boldsymbol{n})$ and $P_{ss}(\boldsymbol{n})$ leads to a positive non-adiabatic entropy production. The adiabatic contribution $\dot{\Sigma}_a$ corresponds to the dissipation of 'house-keeping heat'[20,21], and at steady state matches the entropy flow rate $\dot{\Sigma}_e$. An important property of the previous decomposition is that both contributions are individually positive: $\dot{\Sigma}_a \geq 0$ and $\dot{\Sigma}_{na} \geq 0$[22-25]. Thus, the last inequality and the second line in Eq. (11) imply that the relative entropy $D$ decreases monotonically, and since $D$ is positive by definition, it is a Lyapunov function for the stochastic dynamics.

## Macroscopic limit

In the following we will assume the existence of a scale parameter $\Omega$ controlling the size of the system in question. For example, $\Omega$ can be taken to be the volume $V$ of the solution in well-mixed chemical reaction networks, or the typical value $C$ of capacitance in the case of electronic circuits (see the example below). In addition, we will assume that for large $\Omega$ i) that the typical values of the density $\boldsymbol{x} \equiv \boldsymbol{n}/\Omega$ are intensive, ii) that the internal energy and entropy functions $E(\Omega\boldsymbol{x})$ and $S(\Omega\boldsymbol{x})$ are extensive, and iii) that the transition rates $\lambda_\rho(\Omega\boldsymbol{x})$ are also extensive. Under those conditions, the probability distribution $P_t(\boldsymbol{x})$ satisfies a large deviations (LD) principle[17,26,27]:

$$P_t(\boldsymbol{x}) \asymp e^{-\Omega I_t(\boldsymbol{x})}, \qquad (12)$$

which just means that the limit $I_t(\boldsymbol{x}) \equiv \lim_{\Omega \to \infty} -\log(P_t(\boldsymbol{x}))/\Omega$ is well defined. Then, $I_t(\boldsymbol{x})$ is a positive, time-dependent 'rate function', since it gives the rate at which the probability of fluctuation $\boldsymbol{x}$ decays with the scale. Note that, by Eq. (12), the steady state self-information introduced in Eq. (2) satisfies $\mathcal{I}(\boldsymbol{x}) = \Omega I_{ss}(\boldsymbol{x})$ to dominant order in the macroscopic limit. In other words, the large deviations principle states that the instantaneous self-information $\mathcal{I}_t(\boldsymbol{x}) \equiv -\log(P_t(\boldsymbol{x}))$ is an extensive quantity[26], and we can think of the rate function as the self-information density. Thus, in the following we will consider the ansatz $P_t(\boldsymbol{x}) = e^{-\Omega I_t(\boldsymbol{x})}/Z_t$, with $Z_t \equiv \sum_{\boldsymbol{x}} e^{-\Omega I_t(\boldsymbol{x})}$, as an approximation to the actual time-dependent distribution. This amounts to neglecting sub-extensive contributions to the instantaneous self-information. As explained below, $I_t(\boldsymbol{x})$ takes its minimum value $I_t(\boldsymbol{x}_t) = 0$ at the deterministic trajectory $\boldsymbol{x}_t$, which is equivalent to $P_t(\boldsymbol{x}) = \delta(\boldsymbol{x} - \boldsymbol{x}_t)$ for $\Omega \to \infty$. Plugging the previous ansatz in the master equation of Eq. (5) we note that $\lambda_\rho(\boldsymbol{x} - \boldsymbol{\Delta}_\rho/\Omega) P_t(\boldsymbol{x} - \boldsymbol{\Delta}_\rho/\Omega) \simeq \lambda_\rho(\boldsymbol{x}) P_t(\boldsymbol{x}) e^{\boldsymbol{\Delta}_\rho \cdot \nabla I_t(\boldsymbol{x})}$ to dominant order in $\Omega \to \infty$. Noting also that $\log(Z_t)$ is sub-extensive, it is possible to see that $I_t(\boldsymbol{x})$ evolves according to

$$\partial_t I_t(\boldsymbol{x}) = \sum_\rho \omega_\rho(\boldsymbol{x}) \left[ 1 - e^{\boldsymbol{\Delta}_\rho \cdot \nabla I_t(\boldsymbol{x})} \right], \qquad (13)$$

where $\omega_\rho(\boldsymbol{x}) \equiv \lim_{\Omega\to\infty} \lambda_\rho(\Omega\boldsymbol{x})/\Omega$ are the scaled jump rates[17,28]. In a similar way, in the macroscopic limit the LDB conditions in Eq. (4) take the form

$$\log \frac{\omega_\rho(\boldsymbol{x})}{\omega_{-\rho}(\boldsymbol{x})} = -\beta \left[ \boldsymbol{\Delta}_\rho \cdot \nabla\phi(\boldsymbol{x}) - W_\rho(\boldsymbol{x}) \right], \qquad (14)$$

in terms of the free energy density $\phi(\boldsymbol{x}) \equiv \lim_{\Omega\to\infty} \Phi(\Omega\boldsymbol{x})/\Omega$ (internal energy and entropy densities $\epsilon(\boldsymbol{x})$ and $s(\boldsymbol{x})$ satisfying $\phi(\boldsymbol{x}) = \epsilon(\boldsymbol{x}) - Ts(\boldsymbol{x})$ can be defined in the same way). For the work contributions in Eq. (14) we are abusing notation by writing $W_\rho(\boldsymbol{x}) = \lim_{\Omega\to\infty} W_\rho(\boldsymbol{n} = \Omega\boldsymbol{x})$. Note that we assume that work contributions are intensive. This is justified since they are given by the product of two intensive quantities: a thermodynamic force (for example a potential difference), and the change in a conserved quantity (mass, charge, etc) during a single jump[29]. However, note also that the work rate $\langle \dot{W} \rangle$ will be extensive in general due to the extensivity of the transition rates.

Many classes of systems satisfy the previous scaling assumptions besides the examples already mentioned. Additional examples include non-equilibrium many-body problems like the driven Potts model[30,31], reaction-diffusion models[32,33], and asymmetric exclusion processes[32,34].

From Eq. (12) we see that as $\Omega$ is increased, $P_t(\boldsymbol{x})$ is increasingly localised around the minimum of the rate function $I_t(\boldsymbol{x})$, which is the most probable value. Also, deviations from that typical state are exponentially suppressed in $\Omega$. Thus, the limit $\Omega \to \infty$ is a macroscopic low-noise limit where a deterministic dynamic emerges. In fact, from Eq. (13) one can show that the evolution of the minima $\boldsymbol{x}_t$ of $I_t(\boldsymbol{x})$ is ruled by the closed non-linear differential equations

$$d_t \boldsymbol{x}_t = \boldsymbol{u}(\boldsymbol{x}_t), \text{ with } \boldsymbol{u}(\boldsymbol{x}) \equiv \sum_{\rho>0} i_\rho(\boldsymbol{x}) \boldsymbol{\Delta}_\rho, \qquad (15)$$

where $i_\rho(\boldsymbol{x}) \equiv \omega_\rho(\boldsymbol{x}) - \omega_{-\rho}(\boldsymbol{x})$ are the scaled deterministic currents[17]. The vector field $\boldsymbol{u}(\boldsymbol{x})$ corresponds to the deterministic drift in state space. For chemical reaction networks the dynamical equations in Eq. (15) are the chemical rate equations, while for electronic circuits they are provided by regular circuit analysis.

In the following section we obtain bounds for the steady state rate function $I_{ss}(\boldsymbol{x})$, that according to Eq. (13) satisfies:

$$0 = \sum_\rho \omega_\rho(\boldsymbol{x}) \left[ 1 - e^{\boldsymbol{\Delta}_\rho \cdot \nabla I_{ss}(\boldsymbol{x})} \right]. \qquad (16)$$

## Emergent second law

The positivity of the adiabatic and non-adiabatic contributions to the entropy production, $\dot{\Sigma}_a \geq 0$ and $\dot{\Sigma}_{na} \geq 0$, in addition to the usual second law $\dot{\Sigma} \geq 0$, have been called the 'three faces of the second law'[23]. In[28], the inequality $\dot{\Sigma}_{na} = -k_b d_t D \geq 0$ was put forward as an 'emergent' second law. There, $\mathcal{F} = k_b D$ was interpreted as an alternative non-equilibrium free energy, with a balance equation $d_t \mathcal{F} = \dot{\Sigma}_a - \dot{\Sigma} \leq 0$ (note the analogy with Eq. (9)). Then, the adiabatic contribution $\dot{\Sigma}_a$ was interpreted as an energy input, which at steady state balances the dissipation $\dot{\Sigma}$. Although this point of view is compelling, it is hindered by the fact that there is no clear interpretation of $\dot{\Sigma}_a$ away from the steady state, that would allow to compute this quantity in terms of actual physical currents. In this work we take the other possible road, and investigate the interpretation and consequences of $\dot{\Sigma}_a \geq 0$. We begin by rewriting Eq. (10) using the LDB conditions of Eq. (4) and the definition of $\mathcal{I}$ in Eq. (2), obtaining:

$$\dot{\Sigma}_a = \dot{\Sigma} + k_b d_t \langle \mathcal{I} \rangle - d_t \langle S_{sh} \rangle \geq 0, \qquad (17)$$

where we have defined $\langle \mathcal{I} \rangle = \sum_{\boldsymbol{n}} \mathcal{I}(\boldsymbol{n}) P_t(\boldsymbol{n})$ as the average of the steady state self-information $\mathcal{I}(\boldsymbol{n}) = -\log(P_{ss}(\boldsymbol{n}))$ and $\langle S_{sh} \rangle = -k_b \sum_{\boldsymbol{n}} P_t(\boldsymbol{n}) \log(P_t(\boldsymbol{n}))$ as the Shannon contribution to the system entropy, computed over the instantaneous distribution. Eq. (17) has been already obtained in[22–24], although it was not explicitly written in terms of the self-information $\mathcal{I}$. It is important to note that $\langle S_{sh} \rangle$ is sub-extensive in $\Omega$ (according to Eq. (12), it grows as $\log(\Omega)$), and therefore can be neglected in the macroscopic limit. Thus, changes in average self-information can be bounded by the entropy production, that can in turn be computed or measured in terms of actual energy and entropy flows (see Eqs. (7) and (8)). However, the result in Eq. (17) is not yet in a useful form, since the average $\langle \mathcal{I} \rangle$ does not depend only on $\mathcal{I}(\boldsymbol{n})$, the unknown quantity we are interested in, but also on the instantaneous distribution $P_t(\boldsymbol{n})$, that is also typically unknown. This issue is circumvented in the macroscopic limit, since in that case $P_t(\boldsymbol{x})$ is strongly localised around the deterministic values $\boldsymbol{x}_t$, and therefore $\langle \mathcal{I} \rangle \simeq \Omega I_{ss}(\boldsymbol{x}_t)$ to dominant order in $\Omega \rightarrow \infty$. Thus, in the same limit, Eq. (17) for the adiabatic entropy production rate $\dot{\Sigma}_a$ reduces to

$$\dot{\sigma}_a(\boldsymbol{x}_t) = \dot{\sigma}(\boldsymbol{x}_t) + k_b\, d_t I_{ss}(\boldsymbol{x}_t) \geq 0, \tag{18}$$

where we have defined $\dot{\sigma}(\boldsymbol{x}_t) = \lim_{\Omega \rightarrow \infty} \dot{\Sigma}/\Omega$ as the scaled macroscopic limit of the entropy production rate ($\dot{\sigma}_a(\boldsymbol{x}_t)$ is defined in a similar way). Eq. (18) is a more rigorous version of our central result in Eq. (3), which is obtained by integrating Eq. (18) along deterministic trajectories (satisfying Eq. (15)) and multiplying by the scale factor. It is also useful to write down the first and second laws in the macroscopic limit. The energy balance in Eq. (6) reduces to

$$d_t \epsilon(\boldsymbol{x}_t) = \boldsymbol{u}(\boldsymbol{x}_t) \cdot \nabla \epsilon(\boldsymbol{x}_t) = \dot{w}(\boldsymbol{x}_t) + \dot{q}(\boldsymbol{x}_t), \tag{19}$$

where the scaled heat and work rates for state $\boldsymbol{x}$ are defined as $\dot{q}(\boldsymbol{x}) = \sum_{\rho > 0} i_\rho(\boldsymbol{x}) Q_\rho(\boldsymbol{x})$ and $\dot{w}(\boldsymbol{x}) = \sum_{\rho > 0} i_\rho(\boldsymbol{x}) W_\rho(\boldsymbol{x})$, respectively. Finally, again neglecting the sub-extensive Shannon contribution $\langle S_{sh} \rangle$, the second law in Eq. (7) reduces to

$$\begin{aligned} \dot{\sigma}(\boldsymbol{x}_t) &= -\dot{q}(\boldsymbol{x}_t)/T + d_t s(\boldsymbol{x}_t) \\ &= k_b \sum_{\rho > 0} (\omega_\rho(\boldsymbol{x}_t) - \omega_{-\rho}(\boldsymbol{x}_t)) \log \frac{\omega_\rho(\boldsymbol{x}_t)}{\omega_{-\rho}(\boldsymbol{x}_t)} \geq 0. \end{aligned} \tag{20}$$

## Linear response regime

We will now show that to first order in the work contributions $W_\rho(\boldsymbol{x})$ the inequality in Eq. (18) is saturated. In first place we rewrite Eq. (18) using the macroscopic first and second laws in Eqs. (19) and (20):

$$\dot{\sigma}_a(\boldsymbol{x}_t)/k_b = \boldsymbol{u}(\boldsymbol{x}_t) \cdot \nabla (I_{ss}(\boldsymbol{x}) - \beta \phi(\boldsymbol{x}))|_{\boldsymbol{x}=\boldsymbol{x}_t} + \beta \dot{w}(\boldsymbol{x}_t), \tag{21}$$

where we also used $\phi(\boldsymbol{x}) = \epsilon(\boldsymbol{x}) - Ts(\boldsymbol{x})$ and that $d_t F(\boldsymbol{x}_t) = \boldsymbol{u}(\boldsymbol{x}_t) \cdot \nabla F(\boldsymbol{x}_t)$ for any function $F(\boldsymbol{x})$. Secondly, we note that in detailed-balanced settings (i.e., if $W_\rho(\boldsymbol{x}) = 0 \,\forall\, \rho, \boldsymbol{x}$) the steady state rate function is just $I_{ss}(\boldsymbol{x}) = \beta \phi(\boldsymbol{x})$ (up to a constant), in accordance to the Gibbs distribution (this follows from Eqs. (14) and (16)). Thus, the difference $g(\boldsymbol{x}) \equiv I_{ss}(\boldsymbol{x}) - \beta \phi(\boldsymbol{x})$ appearing in Eq. (18) quantifies the deviations from thermal equilibrium. Expanding Eq. (16) to first order in $W_\rho(\boldsymbol{x})$ and $g(\boldsymbol{x})$, it can be shown that

$$\boldsymbol{u}^{(0)}(\boldsymbol{x}) \cdot \nabla g(\boldsymbol{x}) = -\beta \dot{w}^{(0)}(\boldsymbol{x}) + \mathcal{O}(W_\rho^2), \tag{22}$$

where $\boldsymbol{u}^{(0)}(\boldsymbol{x}) = \sum_{\rho > 0} i_\rho^{(0)}(\boldsymbol{x}) \boldsymbol{\Delta}_\rho$ and $\dot{w}^{(0)}(\boldsymbol{x}) = \sum_{\rho > 0} i_\rho^{(0)}(\boldsymbol{x}) W_\rho(\boldsymbol{x})$ are the lowest-order deterministic drift and work rate, respectively[17]. These are defined in terms of the detailed-balanced deterministic currents $i_\rho^{(0)}(\boldsymbol{x}) = \omega_\rho^{(0)}(\boldsymbol{x}) - \omega_{-\rho}^{(0)}(\boldsymbol{x})$ constructed from the scaled transition rates evaluated at $W_\rho(\boldsymbol{x}) = 0$ that, according to the LDB conditions of Eq. (14),

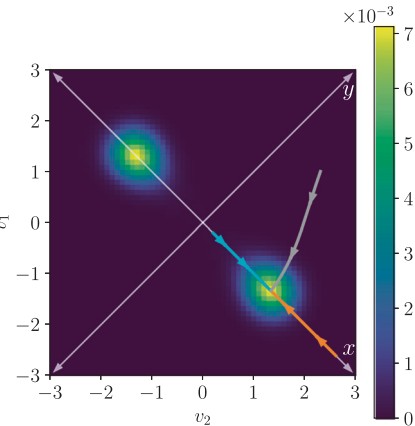

**Fig. 1 | Steady state and deterministic trajectories for the CMOS memory.** Exact steady state distribution for $V_{dd} = 1.4$ and $v_e = 0.1$. The orange, cyan and grey lines show three deterministic trajectories for the same parameters and different initial conditions, with the arrows indicating the direction of motion.

satisfy $\log(\omega_\rho^{(0)}(\boldsymbol{x})/\omega_{-\rho}^{(0)}(\boldsymbol{x})) = -\beta \boldsymbol{\Delta}_\rho \cdot \nabla \phi(\boldsymbol{x})$. Comparing the result of Eq. (22) with Eq. (21), we see that $\dot{\Sigma}_a = 0$ to linear order in $W_\rho(\boldsymbol{x})$. Then, in the linear response regime we can write:

$$\begin{aligned} I_{ss}(\boldsymbol{x}_t) - I_{ss}(\boldsymbol{x}_0) &\simeq -\int_0^t dt'\, \dot{\sigma}^{(0)}(\boldsymbol{x}_{t'})/k_b \\ &= \beta \left[ \phi(\boldsymbol{x}_t) - \phi(\boldsymbol{x}_0) - \int_0^t dt'\, \dot{w}^{(0)}(\boldsymbol{x}_{t'}) \right]. \end{aligned} \tag{23}$$

where the integration is performed along trajectories solving the detailed-balanced deterministic dynamics $d_t \boldsymbol{x}_t = \boldsymbol{u}^{(0)}(\boldsymbol{x}_t)$.

## Example: an electronic memory

In order to illustrate our results we will consider the model of a low-power CMOS memory cell developed in[35,36], that we review in the Supplementary Note 1. This model involves two CMOS inverters connected in a loop, and each inverter is composed of two MOS transistors. There are two degrees of freedom: voltages $v_1$ and $v_2$, that can take values spaced by the elementary voltage $v_e = q_e/C$, where $q_e$ is the positive electron charge and $C$ is a value of capacitance that increases with the scale of the MOS transistors. Thus, in this context the scale parameter can be taken to be $\Omega = V_T/v_e$, where $V_T = k_b T/q_e$ is the thermal voltage. In the following all voltages will be expressed in units of $V_T$. Figure 1 shows a typical steady state distribution in the bistable phase, and three different deterministic trajectories. The logical state of the memory is codified in the sign of the variable $x = v_1 - v_2$. The rate function $I_{ss}(x)$ along the $x$ axis in Fig. 1 can be computed exactly[36]:

$$I_{ss}(x) = x^2 + x V_{dd} + \frac{2n}{n+2} \left[ L(x, V_{dd}) - L(x, -V_{dd}) \right], \tag{24}$$

where $V_{dd}$ is the powering voltage that takes the memory out of thermal equilibrium, $L(x, V_{dd}) = \mathrm{Li}_2(-\exp(V_{dd} + x(1+2/n)))$, and $\mathrm{Li}_2(\cdot)$ is the polylogarithm function of second order. Also, $n \geq 1$ is a parameter that characterizes the transistors (the slope factor), and that will be fixed to $n = 1$ in the following. In the Supplementary Note 1 we show that the rate function in Eq. (24) provides an essentially exact description of the steady state distribution for scale parameters as low as $\Omega = 10$ (i.e., states with at most tens of electrons are appreciably populated). Thus, sub-extensive contributions to the self-information can be safely neglected in this case, even away from the strict macroscopic limit.

In Fig. 2a we show that there is a non-equilibrium transition from a monostable phase into the bistable phase that allows the storage of a

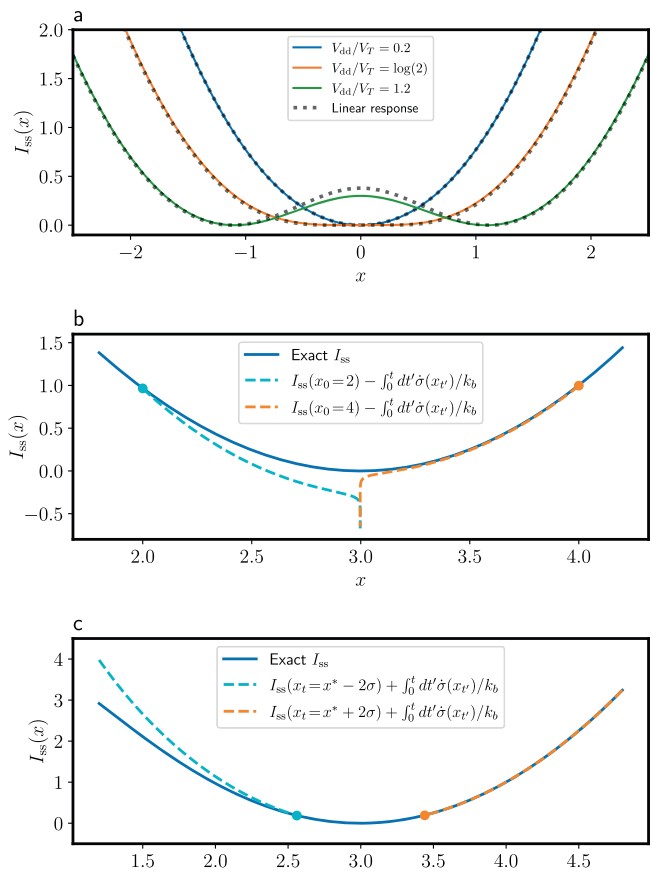

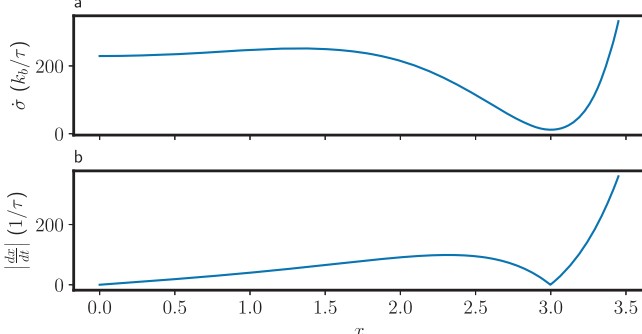

**Fig. 3 | Entropy production rate and speed along deterministic trajectories.** **a** Deterministic entropy production rate $\dot{\sigma}$, and (**b**) deterministic speed $|dx/dt|$ as a function of $x$ for $V_{dd} = 3$ (see the Supplementary Note 1 for the definition of the timescale $\tau$).

**Fig. 2 | Rate function of the CMOS memory. a** Rate function $I_{ss}(x)$ for different values of the powering voltage $V_{dd}$. We also compare $I_{ss}(x)$ with the linear response approximation obtained from Eq. (23) for $V_{dd} = 1.2$. **b** Illustration of the bound to $I(x_t)$ in Eq. (18) for two different deterministic trajectories starting at $x_0 = 2$ and $x_0 = 4$ ($V_{dd} = 3$). **c** Bound to $I_{ss}(x_0)$ for a fixed $x_t$ according to Eq. (18), for $x_t = x^* \pm 2\delta$ ($\delta = 1/\sqrt{2\Omega}$, $\Omega = 10$, and $V_{dd} = 3$).

bit of information, occurring at the critical powering voltage $V_{dd}^* = \ln(2)$. We also compare the exact rate function in Eq. (24) with the linear response approximation obtained from Eq. (23) for different powering voltages. Remarkably, despite it is only expected to be valid close to equilibrium, the linear response approximation captures the transition to bistability and continues to be a reasonable approximation well into the bistable phase. The reason is that in this case the second order non-equilibrium correction vanishes (as can be checked by expanding Eq. (24) in $V_{dd}$), and therefore the linear response approximation is actually valid up to order $V_{dd}^3$.

In Fig. 2b the exact rate function $I_{ss}(x_t)$ along a deterministic trajectory $x_t$ is compared with the lower bound $I_{ss}(x_0) - \int_0^t dt' \dot{\sigma}(x_{t'})/k_b$, for two different trajectories starting at $x_0 = 2$ and $x_0 = 4$, with $y_0 = 0$ in both cases (as shown in Fig. 1, deterministic trajectories initialized in the $x$ axis remain in it). In both cases the trajectory $x_t$ approaches the fixed point $x^* \simeq V_{dd} = 3$. We see that $I_{ss}(x_0) - \int_0^t dt' \dot{\sigma}(x_{t'})/k_b$ is indeed a lower bound to $I_{ss}(x_t)$, in accordance with Eq. (18). Note that this bound diverges when the trajectory approaches the fixed point $x^*$. The reason is that once $x_t \simeq x^*$, the entropy production $\int_0^t dt' \dot{\sigma}(x_{t'})$ just continuously integrates the steady state heat dissipation rate $-\dot{q}(x^*)$ (see Eq. (20)). The linear response approximation avoids this issue since the lowest-order work rate $\dot{w}^{(0)}(x)$ vanishes at the equilibrium fixed point (see Eq. (23)). The divergence can be also avoided by the coarse-graining procedure discussed in the next section. Alternatively, Eq. (18) can be considered an upper bound to $I(x_0)$ for a fixed final point $x_t$. This

is shown in Fig. 2c, for final points $x_t = x^* \pm 2\delta$, where $\delta = 1/\sqrt{2\Omega}$ estimates the variance of the fluctuations around the fixed point.

The fact that in Fig. 2b, c the bounds are much tighter to one side of the fixed point than the other can be traced back to the different speeds at which the fixed point is approached. To show this, in Fig. 3 we have plotted the entropy production rate $\dot{\sigma}$ and the speed $|dx/dt|$ as a function of $x$ for $V_{dd} = 3$. In Fig. 3a we see that $\dot{\sigma}$ is minimized close to the deterministic fixed points (this is a design feature of CMOS devices, in order to minimize the static power consumption, which is however not zero). Also, we see that $\dot{\sigma}$ is actually lower to the left of the fixed point at $x \simeq 3$ than to the right. However, in Fig. 3b we see that the speed $dx/dt$ at which the fixed point is approached is also lower to the left side, resulting in a larger total entropy production $\int_0^t dt' \dot{\sigma}(x_{t'}) = \int_{x_0}^{x_t} dx \dot{\sigma}(x)/(dx/dt)$, and a looser bound in Fig. 2b, c.

## Tightening the bound

Our main result in Eq. (18) allows to obtain information about the steady state fluctuations by just measuring the physical entropy production along deterministic trajectories. However, if we consider the mathematical problem of bounding the steady state fluctuations given the transition rates $\lambda_\rho(\boldsymbol{n})$, then the full power of our result is achieved by considering a 'coarse-grained' entropy production that is in general a lower bound to the actual physical one, as we now explain. In first place we notice that different sets of transition rates $\{\lambda_\rho(\boldsymbol{n})\}$ might lead to the same master equation (Eq. (5)) and consequently the same steady state and emerging deterministic dynamics (Eq. (15)), while giving rise to different entropy production rates (Eq. (7)). This is due to the fact that the entropy production depends on how a given stochastic dynamics is split into thermodynamically distinct processes, each of them satisfying a different LDB condition. In particular, the entropy production is not invariant under coarse-graining of the transition rates[23,37]. Specifically, consider that we lump together all transitions going from state $\boldsymbol{n}$ to state $\boldsymbol{n} + \tilde{\boldsymbol{\Delta}}_\rho$ into a single transition with rate

$$\tilde{\lambda}_\rho(\boldsymbol{n}) \equiv \sum_{\rho/\boldsymbol{\Delta}_\rho = \tilde{\boldsymbol{\Delta}}_\rho} \lambda_\rho(\boldsymbol{n}). \tag{25}$$

In this way, the jump vectors $\{\tilde{\boldsymbol{\Delta}}_\rho\}$ associated to the coarse-grained rates $\tilde{\lambda}_\rho(\boldsymbol{n})$ are all distinct ($\tilde{\boldsymbol{\Delta}}_\rho \neq \tilde{\boldsymbol{\Delta}}_{\rho'}$ if $\rho \neq \rho'$). It is easy to check that rates $\{\tilde{\lambda}_\rho(\boldsymbol{n})\}$ and $\{\lambda_\rho(\boldsymbol{n})\}$ lead to the same master equation. However, the entropy production rate $\dot{\Pi}$ corresponding to the rates $\{\tilde{\lambda}_\rho(\boldsymbol{n})\}$ is always a lower bound of the original one[23]:

$$\dot{\Pi} = \frac{k_b}{2} \sum_{\rho,\boldsymbol{n}} (\tilde{j}_\rho(\boldsymbol{n}) - \tilde{j}_{-\rho}(\boldsymbol{n} + \tilde{\boldsymbol{\Delta}}_\rho)) \log \frac{\tilde{j}_\rho(\boldsymbol{n})}{\tilde{j}_{-\rho}(\boldsymbol{n} + \tilde{\boldsymbol{\Delta}}_\rho)} \leq \dot{\Sigma} \tag{26}$$

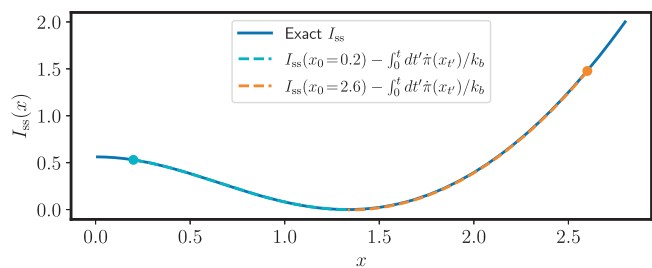

**Fig. 4 | Coarse-grained bound for the CMOS memory.** Illustration of the bound to $I(x_t)$ from Eq. (18) in terms of the entropy production rate $\dot{\pi}$ for the cyan and orange trajectories in Fig. 1, starting respectively at $x_0 = 0.2$ and $x_0 = 2.6$ ($V_{dd} = 1.4$).

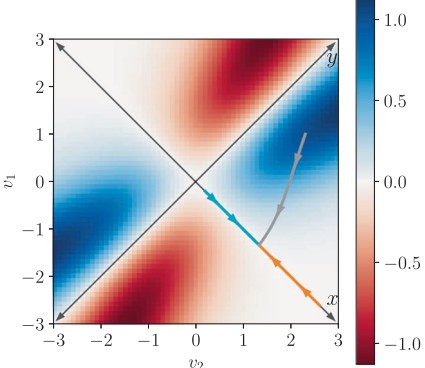

**Fig. 5 | Vorticity and deterministic trajectories.** Vorticity $f(x)$ of the vector field with components $\tilde{\sigma}_\rho(x) \equiv \log(\tilde{\omega}_\rho(x)/\tilde{\omega}_{-\rho}(x))$, $\rho = 1, 2$, for the CMOS memory model ($V_{dd} = 1.4$ and $v_e = 0.1$, as in Fig. 1). The orange, cyan and grey lines correspond to the same deterministic trajectories of deterministic trajectories of Fig. 1.

An important property of $\dot{\Pi}$ is that, whenever the emerging deterministic dynamics has fixed point attractors, its macroscopic version $\dot{\pi}(x_t) \equiv \lim_{\Omega \to \infty} \dot{\Pi}/\Omega = \sum_{\rho > 0}(\tilde{\omega}_\rho(x_t) - \tilde{\omega}_{-\rho}(x_t)) \log(\tilde{\omega}_\rho(x_t)/\tilde{\omega}_{-\rho}(x_t))$ vanishes at the fixed points (see the Supplementary Note 2 for a detailed discussion). Thus, the bound obtained using the entropy production rate $\dot{\pi}$ is free from the divergence as the fixed point is approached. In the case of the CMOS memory, this is shown in Fig. 4 for the two trajectories along the $x$ axis of Fig. 1. We see that not only the divergence is avoided, but that the bound matches the exact rate function (see below).

In analogy with Eq. (14), whenever the log-ratio $\tilde{\sigma}_\rho(x) \equiv \log(\tilde{\omega}_\rho(x)/\tilde{\omega}_{-\rho}(x))$ of the scaled coarse-grained rates can be expressed as the gradient of a state function $-\tilde{\phi}(x)$ along the direction $\tilde{\Delta}_\rho$, the steady state rate function will be just $I_{ss}(x) = \tilde{\phi}(x)$ (up to a constant). In that case, the system can be considered to be at equilibrium at the coarse-grained level. Under some conditions, one can easily test if $\tilde{\sigma}_\rho(x)$ derives from a gradient by just checking if the generalized curl $f_{\rho,\rho'}(x) \equiv \partial_{x_\rho}\tilde{\sigma}_{\rho'}(x) - \partial_{x_{\rho'}}\tilde{\sigma}_\rho(x)$ vanishes for all $\rho$, $\rho'$, and $x$ (see the Supplementary Note 2). If that is the case, then the bound in Eq. (18) in terms of the entropy production rate $\dot{\pi}$ is saturated, and one can fully reconstruct the steady-state rate function in terms of the deterministic dynamics.

In the CMOS memory the state space is two dimensional and therefore there is a single curl component $f(x)$, or vorticity. In Fig. 5 we show the vorticity $f(x)$ for the same parameters as in Fig. 1. We see that the model is genuinely non-equilibrium even at the coarse-grained level. However, we also see that the vorticity vanishes at the $x$ and $y$ axes, which explains why the bound in Fig. 4 matches the exact rate function. Finally, we study how the bound in terms of the coarse-grained entropy production rate $\dot{\pi}$ performs for the gray trajectory in Figs. 1 and 5, that starts in and goes through areas of non-vanishing

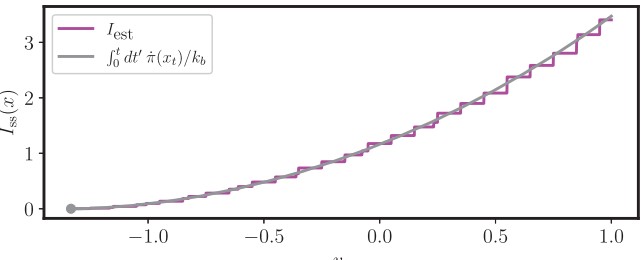

**Fig. 6 | Coarse-grained bound for non-zero vorticity.** Comparison of the upper bound to $I_{ss}(x_0)$ in terms of the coarse-grained entropy production rate $\dot{\pi}$ for the points in the gray trajectory in Figs. 1 and 5 with the function $I_{est}(x) = -\log(P_{ss}(x))/\Omega$ obtained from the exact steady state distribution.

vorticity, and thus genuinely out of equilibrium. Since in this case we do not have the exact rate function to compare with, we compare the bound with the estimation of the rate function $I_{est}(x) = -\log(P_{ss}(x))/\Omega$ obtained from the exact steady state distribution. $I_{est}(x)$ is only defined for the discrete set of voltages $v_1$ and $v_2$ that are a multiple of the elementary voltage $v_e$, and in the comparison we choose the closest values of $v_1$ and $v_2$ to the continuous deterministic trajectory $x_t$, obtaining a stepwise function. The results are shown in Fig. 6. We see that even in this case our bound provides an accurate approximation of the probability distribution. This shows that the emergent second law in terms of the coarse-grained entropy production has the potential to offer a deterministic alternative to Gillespie simulations and spectral methods. As an example, we provide in the Supplementary Note 3 a full reconstruction of the steady state distribution of the CMOS memory.

It is important to note that the bound given by the emergent second law in terms of the mathematical entropy production in Eq. (26) can be applied to any Markov jump process displaying a macroscopic limit, irrespective of whether it represents a thermodynamically consistent stochastic dynamics or not (the only restriction is that for each jump the reverse jump should also be possible). For example, our results can be relevant in stochastic population[38,39] or gene expression[40,41] models.

## Discussion

For systems accepting a description in terms of Markov jump processes, our results unveil a fundamental connection between the deterministic dynamics that emerges in a macroscopic limit and the non-equilibrium fluctuations at steady state. This is given by an inequality that can be interpreted as an emergent second law. In fact, it is a tighter version of the usual second law, that is saturated in the linear response regime. As shown in the Supplementary Note 4, the emergent second law can be alternatively understood as a generalized fluctuation-dissipation relation. The practical value of our result lies in the fact that the probability of non-equilibrium fluctuations is hard to evaluate, while the deterministic dynamics is directly given by standard methods. The corresponding linear response theory, working at the level of the rate function, was shown in[17] to be highly accurate in some model systems, with a regime of validity beyond that of usual linear response theories. Our result can also be employed in combination with numerical or experimental approaches: once normal or moderately rare fluctuations have been sampled and characterized, Eq. (3) can be used to bound the probability of very rare fluctuations, that otherwise would require extremely long simulation times. In addition, the refinement of our result by a coarse-graining procedure leads to novel numerical techniques to compute non-equilibrium distributions, that only rely on the deterministic dynamics and thus offer an alternative to stochastic simulations or spectral methods.

As a final comment, we note that the extensivity assumptions defining the macroscopic limit we have considered are a very natural extension of the usual notion of extensivity in equilibrium thermodynamics. The only additional requirements, besides the extensivity of the free energy associated to each state, are the extensivity of the jump rates between different states, and the intensivity of the work contributions. These natural requirements lead to an extensive non-equilibrium self-information, whose dominant contribution is constrained by our emergent second law, and that at equilibrium reduces to the regular extensive free energy.

## Methods

As explained in the main text, the emergent second law can be obtained from the fact that the adiabatic contribution to the entropy production rate is positive. The proof of this fact can be found in the mentioned references. The details of the model in the example are given in the Supplementary Note 1. The properties of the coarse grained entropy production are discussed in the Supplementary Note 2. In the Supplementary Note 3 we show how the emergent second law can provide an alternative to stochastic simulations or spectral methods. Finally, in the Supplementary Note 4 we make explicit the connection between the emergent second law derived here and fluctuation-dissipation relations.

## Data availability

All the data shown in the figures can be easily reproduced from the given information, and is also available at the Git repository https://github.com/nfreitas/emergent-second-law.git.

## Code availability

The scripts used to generate the data shown in the plots are available at the Git repository: https://github.com/nfreitas/emergent-second-law.git.

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

## Acknowledgements
We acknowledge funding from the INTER project "TheCirco" (INTER/FNRS/20/15074473) and CORE project "NTEC" (C19/MS/13664907), funded by the Fonds National de la Recherche (FNR, Luxembourg), and from the European Research Council, project NanoThermo (ERC-2015-CoGAgreement No. 681456). The authors applied a Creative Commons Attribution 4.0 International (CC BY 4.0) license to any Author Accepted Manuscript version arising from this submission.

## Author contributions
J.N.F. and M.E. contributed equally to all aspects of this work.

## Competing interests
The authors declare no competing interests.
