## [Peer Review File · Nature Communications]

Report on NCOMMS-22-04475-v1

Dear editor,

I have read with attention the manuscript entitled “Emergent Second Law for Non-Equilibrium Steady States” by Freitas and Esposito and submitted for publication in Nature Communications.

After briefly summarising what I believe are the paper’s findings, I shall give my comments and recommendation.

The paper extends a series of general results within the formalism of stochastic thermodynamics to systems where the stochastic variables become extensive in the thermodynamic limit and where the stochastic dynamics becomes effectively that of intensive scale-free variables following a macroscopic deterministic dynamics. Such systems may be chemical reaction networks or electric circuits, the latter being considered for application at the end of the article. For such systems the formalism yields a second-law-like equation which is more restrictive than the traditional second law of thermodynamics i.e. it provides a lower bound to the entropy production or, equivalently, relates the changes in the so-called ‘self-information’ for systems in steady state regimes to the entropy production.

The manuscript is well and clearly written and its intended scope surely should be of interest for the wider readership of Nature Communications.

Apart from some small clarifications I believe should be given (which I shall mention below), the derivations are valid and the obtained results are stated carefully enough for them not be misleading or over-generalised.

The idea of using the thermodynamic limit to circumvent the fact that the time-dependent probability for the system’s state is rarely known, while the macroscopic deterministic equations often are, does solve a thorny problem when looking at very large system sizes.

That being said, the submitted work does come from a series of previous works from the authors (some of them as recent as 2021) which already lay out most of the results and derivations presented in the paper. Eq. (18) for example, from which the main result follows for macroscopic variables was already derived in Ref. [28]. As a result, although somewhat significant, the main result of the paper appears as an application of Eq. (18) (which is not original to this paper) to some specific regime where calculations are tractable and, therefore, does not come across as very original as such given the existing body of work on the subject.

For this reason, I believe the submitted work does not meet the originality and very strong significance manuscripts should satisfy to warrant publication in Nature Communications. I would recommend publication in a more specialised journal.

Just a couple of minor comments:

1. At the start of section II, the Markov model being employed is introduced: state space and rates are introduced and the quantity ρ representing the magnitude of the jumps as well but then the quantity Δ_ρ is actually not defined; while it appears everywhere in the subsequent derivations. Of course, we may guess what these are but I believe that these should be clarified.
2. There is a typo on page 5 right column ‘was show in [28]’.

Over the last 20 years, a major development in non-equilibrium statistical physics has dealt with proving various forms and refinements of the second law typically assuming an underlying stochastic dynamics. Here, Freitas and Esposito derive a second law in the macroscopic limit that is claimed to be stronger than the usual one. The main inequality, eq. 3, relates entropy production along the macroscopic deterministic dynamics with the ratio of probabilities for observing the corresponding initial and final microstates.

Since the former is easily accessible, the inequality bounds this probability ratio. While they build on their previous work published in NJP 2021 and, for the illustrative example, PRX 2021, the new results are a timely and potentially significant step that deserves publication in a high profile journal, in principle.

My main problem with the presentation is that I find some of the notation unclear which prevented me from following the derivation in sufficient detail.

1) In Sect. II, I missed a precise definition of the Σ that occurs in eq. 3. The $\dot{\Sigma}$ in eq. 8 is the ordinary total entropy production from stochastic thermodynamics and thus presumably not the time derivative of the one from eq. 3.

2) What is the relation between eq. 18 and eq. 3; is the former the time-derivative of the latter? In eq. 3, one seems to be free to choose any initial and final microstates x_0 and x_t . Eq. 18 refers to an ensemble average.

3) At the end of the conclusions, it is claimed that eq. 3 can be used to bound the probability of rare fluctuations after moderate fluctuations have been sampled. Why is it necessary to sample moderate fluctuations? Isn't a solution of the deterministic dynamics sufficient to apply eq. 3?

Minor issues:

4) Eq. 25, the function $L(x, V_{dd})$ is actually a function of x and V_{dd}/V_T

5) Do the authors have intuition why the linear response approximation captures the transition to bistability, which is a feature characterized on page 5 as "remarkable"?

6) It's nice to see a reference to Einstein's original work, Ref. 4. However, the nouns should then be capitalized properly.

In summary, this is a potentially significant paper that could be of interest to a broad community working on the foundations and application of non-equilibrium statistical physics. However, the issues raised above must be clarified before I can reach a final recommendation.

Reviewer #3 (Remarks to the Author):

“Emergent second law for non-equilibrium steady states”

Nahuel Freitas and Massimiliano Esposito

In this manuscript, the authors recapitulate a derivation of the second-law-like relation (18), and take the macroscopic limit where dynamics become deterministic, giving what the authors dub an emergent second law (3), involving the change in self-information and entropy production. They show that close to equilibrium this becomes an equality. They illustrate these ideas in a CMOS memory cell.

I quite like this paper and found it an interesting and thought-provoking read. The paper is admirably clear in its writing: the mathematical manipulations are for the most part (see a limited number of counterexamples below) clearly described, and the work is contextualized well. Having said this, I would say that the manuscript seems pitched at a relatively specialized statistical-physics audience, rather than the broader science audience of Nature Communications.

The idea of deterministic dynamics giving access to otherwise inaccessible quantities is interesting, but my instinct is that the appeal to the macroscopic limit severely limits the applicability of these ideas to the kinds of systems many in the field have in mind. In particular, while the authors' language seems to indicate that these results hold for “open systems displaying a macroscopic limit where a deterministic dynamics emerges” (i.e., for systems that have a macroscopic limit that is deterministic, the results hold away from the macroscopic limit), as far as I can discern the results only hold in that macroscopic limit.

So in conclusion I do not yet see a strong case for the general accessibility and wide impact that Nature Communications demands; instead I feel this would be a strong contribution to a more specialized journal. But I am happy to hear a rebuttal from the authors about the widespread utility of these results, with some effort to further emphasize the motivation and outcomes for a general audience.

Below I add much more subsidiary points for the authors' consideration:

OPTIONAL DATA ANALYSIS SUGGESTIONS:

* Fig. 1a: why not display the linear-response curves for the other example V 's?

* Fig. 2: add (c) that shows $\Sigma\text{-dot}/|dx/dt|$, so we can see the asymmetry more explicitly?

Highlight domain of Fig. 2 plots that is shown in Fig. 1c?

MORE EXPLANATION WOULD HELP

* Provide more explanation about how to get Eqs. (14), (15), and (22)

POSSIBLE TYPOS

* p4 top left: "an useful form" -> "a useful form"

* Fig. S1: "transistors" -> "transistor"

* pS2 middle: "finite number of dimensions" -> "finite number of states" ?

* pS2 bottom: Figure reference is missing the number.

* pS2 bottom: "change of variables in the functions $a(., .)$ and $b(., .)$ is implicit": should be "function $i(.,.)$ " instead of "functions $a(.,.)$ and $b(.,.)$ " ?

Reply NCOMMS-22-04475-v1

April 2022

1 Summary of changes

All relevant changes in the paper are marked in blue, except an entire new section that was added. We list below the most important changes:

- We added a few sentences at the end of the second paragraph of the introduction in order to clarify the regime of validity of Eq. (3), in response to a doubt raised by the second reviewer.
- We added details about the derivations of Eqs. (13), (14), and (21), in response to the third reviewer.
- We added an inconsequential missing term in Eq. (17), and a better explanation of why it can be neglected.
- We added a figure to the explanation of the example, showing the full steady state of the model as well as different deterministic trajectories.
- We added an explanation about the accuracy of the linear response approximation for the example, in response to the second reviewer.
- We added a new section with original results showing that the bound given by our main result can be made tight via a general coarse-graining procedure. As explained below, this should clear any doubt about the relevance and potential impact of our result.

2 Explanation of the new results

In the previous version of the manuscript we provided a bound to the rate function of steady-state non-equilibrium fluctuations in terms of the entropy production along deterministic trajectories. That entropy production is the actual physical one, that could for example be measured as dissipated heat in an experiment. This is a novel and relevant physical result in itself. However, we have now noticed that one can exploit the fact that different physical models, with different entropy production rates, can share the same stochastic dynamics and therefore the same steady state. One way to generate equivalent models

with minimal entropy production is to ‘coarse-grain’ transition rates by lumping together those that correspond to the same jump in state space (even if they are caused by different physical processes). In this way one can define a mathematical entropy production that provides a lower bound to the actual physical one, and leads to a tighter bound of the steady-state rate function. As we show, when our emergent second law is expressed in terms of this mathematical entropy production, it becomes tight near the fixed points for models in which the emergent deterministic dynamics have fixed point attractors. In fact, for the CMOS memory example, the bound becomes so accurate that it can be used to reconstruct the full non-equilibrium distribution. Thus, our improved result provides a powerful and original deterministic alternative to widespread stochastic Gillespie simulations, or spectral methods. Furthermore, it can be applied to Markov jump processes that have no relation to thermodynamics at all, like models of stochastic population dynamics or gene regulatory networks.

3 Comments from the first reviewer

1) The idea of using the thermodynamic limit to circumvent the fact that the time-dependent probability for the system’s state is rarely known, while the macroscopic deterministic equations often are, does solve a thorny problem when looking at very large system sizes.

That being said, the submitted work does come from a series of previous works from the authors (some of them as recent as 2021) which already lay out most of the results and derivations presented in the paper. Eq. (18) for example, from which the main result follows for macroscopic variables was already derived in Ref. [28]. As a result, although somewhat significant, the main result of the paper appears as an application of Eq. (18) (which is not original to this paper) to some specific regime where calculations are tractable and, therefore, does not come across as very original as such given the existing body of work on the subject.

For this reason, I believe the submitted work does not meet the originality and very strong significance manuscripts should satisfy to warrant publication in Nature Communications. I would recommend publication in a more specialised journal.

Some clarifications are in order. First, it is not true that Eq. (18) (now Eq. (17)) is derived in Ref. [28] (now Ref. [17], after a correction). As we mentioned in the paragraph below it, Eq. (17) and, more generally, the adiabatic/non-adiabatic decomposition, were already obtained much earlier in references [22-24]. It is important to note that although that formal result was known for 15 years, it didn’t have any practical application since its evaluation requires to know the instantaneous probability distribution. The first original contribution of our work is to show how Eq. (17) reduces to an useful form when the macroscopic limit is taken. This is a non-trivial result with far reaching consequences.

Secondly, what is actually shown in our recent work in Ref. [17] is Eq. (23) in the new manuscript. Thus, that work only deals with the linear response analysis, showing that close to equilibrium the steady state rate function can be obtained from the deterministic entropy production. The fact that this corresponds exactly to a vanishing *adiabatic* entropy production (that is, to the saturation of the inequality in Eq. (17)) is a completely independent claim and the second original contribution of the present work.

We hope that the previous clarifications address the concerns about the ‘originality and very strong significance’ expected from an Nature Communications article. In addition, we have now added a new section with original results showing that the bound provided by our emergent second law can be made tight via a coarse graining procedure (see explanation in Section 2 of this document).

2) At the start of section II, the Markov model being employed is introduced: state space and rates are introduced and the quantity ρ representing the magnitude of the jumps as well but then the quantity Δ_ρ is actually not defined; while it appears everywhere in the subsequent derivations. Of course, we may guess what these are but I believe that these should be clarified.

We thank the reviewer for the suggestion, that we implemented.

3) There is a typo on page 5 right column ‘was show in [28]’.

We corrected it.

4 Comments from the second reviewer

1) In Sect. II, I missed a precise definition of the Σ that occurs in eq. 3. The $\dot{\Sigma}$ in eq. 8 is the ordinary total entropy production from stochastic thermodynamics and thus presumably not the time derivative of the one from eq. 3.

$\dot{\Sigma}$ is indeed the ordinary total entropy production from stochastic thermodynamics as defined in Eq. (8) (now Eq. (7)). The quantity Σ appearing in Eq. (3) is just $\Sigma = \int_0^t \dot{\Sigma}(x'_t) dt'$, but evaluated in the macroscopic limit (that is, using Eq. (20) in the updated manuscript).

This question might be related to the fact that, for the sake of simplicity and clarity, we have decided to explain the main features of our result as straightforwardly as possible in the introduction. Therefore, the macroscopic limit was only mentioned, but not properly defined. Now we have added two sentences at the end of the second paragraph of the introduction to clarify the fact that Eq. (3) is strictly valid in the macroscopic limit. However, as explained in detail to the third reviewer and also in the manuscript, our results are also valid for micro and mesoscopic systems if sub-extensive contributions to the self-information can be neglected. This usually becomes an excellent approximation very quickly as one increases the scale parameter, as we exemplify in

the CMOS memory (see the Supplementary Material).

2) *What is the relation between eq. 18 and eq. 3; is the former the time-derivative of the latter? In eq. 3, one seems to be free to choose any initial and final microstates x_0 and x_t . Eq. 18 refers to an ensemble average.*

In Eq. (3) one is only free to choose x_0 , since x_t is given by x_0 and the deterministic dynamics (alternative, one can fix x_t but the values of x_0 must be compatible with the deterministic dynamics). The situation is analogous in Eq. (18) (now Eq. (17)): one is free to choose any initial distribution $P_0(n)$, and the averages are computed over the time-dependent distribution $P_t(n)$, which is obtained from $P_0(n)$ and the master equation in Eq. (6) (now Eq. (5)). Thus, Eq. (3) is just Eq. (17) evaluated in the macroscopic limit and integrated in time. We have now changed the notation in order to make the macroscopic limit more clear.

In relation to this comment, we notice that there was a term missing in Eq. (17), that we corrected now. It was just the Shannon contribution to the system entropy that can be neglected in the macroscopic limit, since it is sub-extensive, and therefore none of the final results were affected. This is more clearly explained now.

3) *At the end of the conclusions, it is claimed that eq. 3 can be used to bound the probability of rare fluctuations after moderate fluctuations have been sampled. Why is it necessary to sample moderate fluctuations? Isn't a solution of the deterministic dynamics sufficient to apply eq. 3?*

Yes, the deterministic dynamics is sufficient to evaluate the bound in Eq. (3), but as explained in the example the bound diverges when the deterministic fixed point is approached. Also, Eq. (3), only allows to bound $I(x_0)$ or $I(x_t)$ if the other one is known. Sampling normal fluctuations allows to estimate the rate function $I(x_t)$ in the neighborhood of the fixed point, and therefore one does not need to let x_t approach the fixed point, avoiding the divergence.

In relation to this question, we note that we have now added a new section explaining a coarse-graining procedure that allows to avoid the divergence without the need to sample normal fluctuations. Also, the bound improved in this way becomes tight near the fixed point. Therefore, the deterministic dynamics is actually enough to reconstruct the rate function near the fixed point (see the section 2 of this document).

4) *Eq. 25, the function $L(x, V_{dd})$ is actually a function of x and V_{dd}/V_T*

We now express all voltages in units of the thermal voltage V_T .

5) *Do the authors have intuition why the linear response approximation captures the transition to bistability, which is a feature characterized on page 5 as "remarkable"?*

One possible intuition comes from the exact solution of the rate function in Eq. (25). Since it is an odd function of V_{dd} , its expansion in that parameter lacks

a second order correction, so the linear response approximation is valid up to order V_{dd}^3 , instead of up to order V_{dd}^2 as in the general case. This is mentioned now.

6) *It's nice to see a reference to Einstein's original work, Ref. 4. However, the nouns should then be capitalized properly.*

We corrected it.

5 Comments from the third reviewer

1) I quite like this paper and found it an interesting and thought-provoking read. The paper is admirably clear in its writing: the mathematical manipulations are for the most part (see a limited number of counterexamples below) clearly described, and the work is contextualized well. Having said this, I would say that the manuscript seems pitched at a relatively specialized statistical-physics audience, rather than the broader science audience of Nature Communications.

We thank the reviewer for the positive evaluation of the article. Regarding its accessibility to a broader audience, in our opinion the main result is introduced in the first part of the article as non-technically as possible, using examples from two fields to which our results can be immediately useful. Also, while the precise statement and derivation of our results is unavoidably technical, we have put great effort in making the article as self-contained as possible, requiring from the reader the less amount of previous knowledge about stochastic thermodynamics (the only non-trivial element in the derivation that is not self-contained in the article is the positivity of the adiabatic and non-adiabatic contributions to the entropy production, which is just stated). We have now added a new figure about the application example, in order to clarify the relationship between the steady state distributions we are interested in and the emerging deterministic dynamics one has access to. We have also extended, as suggested by the reviewer, the explanation about how to get to Eqs. (14), (15) and (22).

The idea of deterministic dynamics giving access to otherwise inaccessible quantities is interesting, but my instinct is that the appeal to the macroscopic limit severely limits the applicability of these ideas to the kinds of systems many in the field have in mind. In particular, while the authors' language seems to indicate that these results hold for "open systems displaying a macroscopic limit where a deterministic dynamics emerges" (i.e., for systems that have a macroscopic limit that is deterministic, the results hold away from the macroscopic limit), as far as I can discern the results only hold in that macroscopic limit.

Indeed, the emergent second law we proved is an statement about the steady-state rate function, which is an object defined in the strict macroscopic limit $\Omega \rightarrow \infty$. However, it is very important to realize that the results we obtained in that way can be applied for finite values of the scale parameter Ω because sub-

extensive contributions to the self-information become very quickly negligible and neglecting them leads to extremely accurate approximations. To show this, we had provided in the Supplementary Material (Fig. 2) a comparison between the exact steady state distribution of the CMOS memory and the distribution reconstructed from the rate function and a scale parameter of only $\Omega = 10$ (or equivalently, $v_e = 0.1$, i.e, only tens of electrons are involved). Our results are thus very relevant in regimes that cannot be considered macroscopic. This is also clear now in the newly added Fig. 6 and the last section of the supplementary material. In relation to this point we have now expanded the discussion below Eq. (12), where the large deviations principle is introduced, and below Eq. (24), where the rate function of the CMOS memory is given.

So in conclusion I do not yet see a strong case for the general accessibility and wide impact that Nature Communications demands; instead I feel this would be a strong contribution to a more specialized journal. But I am happy to hear a rebuttal from the authors about the widespread utility of these results, with some effort to further emphasize the motivation and outcomes for a general audience.

In first place, as explained in the previous point, we stress that our results are not restricted to the strict macroscopic limit. Secondly, we have now added a new section where we explain how our bound to the rate function can be made tight by considering an effective entropy production (see explanation in Section 2 of this document). We also mention that our results are relevant for many different fields. To motivate our work we have focused on non-linear electronic circuits and chemical reactions networks, which are already a large family of models that can display very complex phenomena. However, our results can be directly applied to other interacting many-body problems like the driven Potts model, or spatially extended problems like asymmetric exclusion processes or reaction-diffusion models. This is now mentioned in the manuscript. Finally, the newly added improvement of our bound, based on a mathematical definition of the entropy production rate, can be applied to any Markov jump processes displaying a macroscopic limit, even if it is not thermodynamically consistent or if it lacks any connection to thermodynamics. Thus, our results might also be relevant for stochastic population models or stochastic gene expression.

2) *Fig. 1a: why not display the linear-response curves for the other example V 's?*

We have followed this suggestion and added those curves.

3) *Fig. 2: add (c) that shows $\dot{\Sigma}/|dx/dt|$, so we can see the asymmetry more explicitly? Highlight domain of Fig. 2 plots that is shown in Fig. 1c?*

Now we show only half of the original curves, since they were symmetric, making better use of the available space.

4) *Provide more explanation about how to get Eqs. (14), (15), and (22)*

We have added details in the paragraph between Eqs. (13) and (14) that should

make more clear how to obtain Eqs. (14) and (15). We have also added details about the derivation of Eq. (22).

5) *POSSIBLE TYPOS*

* *p4 top left*: “an useful form” – > “a useful form”

* *Fig. S1*: “transistors” – > “transistor”

* *pS2 middle*: “finite number of dimensions” – > “finite number of states”
?

* *pS2 bottom*: *Figure reference is missing the number.*

* *pS2 bottom*: “change of variables in the functions $a(., .)$ and $b(., .)$ is implicit”: should be “function $i(.,.)$ ” instead of “functions $a(.,.)$ and $b(.,.)$ ”
?

We thank the reviewer for noticing these typos, that we corrected.

REVIEWERS' COMMENTS

Reviewer #1 (Remarks to the Author):

I have read with attention the reply/rebuttal from the authors to the referees and the amended version of the manuscript.

I believe that the changes and clarifications made to the manuscript improve its rigour, accessibility and appeal to a wider audience.

The newly added paragraphs, figures and discussions especially in the Example section, do contribute to making the findings more impactful and original.

I am happy to recommend publication of the manuscript in Nature Communication.

Reviewer #2 (Remarks to the Author):

I have read all reports, the rebuttal letter and the revised manuscript. The paper has gained substantially in clarity and also in interest to a wider community through the revision process. I am happy to recommend publication in Nat Comm as now is.

Reviewer #3 (Remarks to the Author):

The authors have helped me better understand the appeal of their findings and the applicability of their findings. I still feel that the presentation will be hard to follow for all but a specialized audience, and so have moderate concerns about its broad accessibility. But all in all I am happy to give my (qualified) endorsement for publication in Nature Communications.

I note that the authors should provide a data availability statement (it was unchecked on the Editorial Checklist and I could find none in the manuscript).

POSSIBLE TYPOS

p3 mid right: "difussion" -> "diffusion"

Fig. 4 caption: "stating" -> "starting"

p7 mid right: "an stepwise" -> "a stepwise"

SI Ic: "vorticty"

Fig. S3 is missing a/b/c labels

Fig. S3 caption and pS5 top: "same than" -> "same as"

REPLIES FOR NCOMMS-22-04475B

- Reviewer #1 (Remarks to the Author):

I have read with attention the reply/rebuttal from the authors to the referees and the amended version of the manuscript. I believe that the changes and clarifications made to the manuscript improve its rigour, accessibility and appeal to a wider audience. The newly added paragraphs, figures and discussions especially in the Example section, do contribute to making the findings more impactful and original.

I am happy to recommend publication of the manuscript in Nature Communication.

REPLY: We thank the reviewer for the provided feedback and the positive evaluation of our manuscript.

- Reviewer #2 (Remarks to the Author):

I have read all reports, the rebuttal letter and the revised manuscript. The paper has gained substantially in clarity and also in interest to a wider community through the revision process. I am happy to recommend publication in Nat Comm as now is.

REPLY: We thank the reviewer for the provided feedback and the positive evaluation of our manuscript.

- Reviewer #3 (Remarks to the Author):

The authors have helped me better understand the appeal of their findings and the applicability of their findings. I still feel that the presentation will be hard to follow for all but a specialized audience, and so have moderate concerns about its broad accessibility. But all in all I am happy to give my (qualified) endorsement for publication in Nature Communications.

I note that the authors should provide a data availability statement (it was unchecked on the Editorial Checklist and I could find none in the manuscript).

POSSIBLE TYPOS

p3 mid right: "difussion" -> "diffusion"

Fig. 4 caption: "stating" -> "starting"

p7 mid right: "an stepwise" -> "a stepwise"

SI Ic: "vorticty"

Fig. S3 is missing a/b/c labels

Fig. S3 caption and pS5 top: "same than" -> "same as"

REPLY: We thank the reviewer for the provided feedback and the positive evaluation of our manuscript. We have corrected the mentioned typos and added the data availability statement.